# Pregnancy-related interventions in mothers at risk for gestational diabetes in Asian India and low and middle-income countries (PRIMORDIAL study): protocol for a randomised controlled trial

Senthil K Vasan [1,2] Modou Jobe,[3] Jiji Mathews,[4] Fatoumata Cole,[5] Swathi Rathore,[4] Ousman Jarjou,[3] Dylan Thompson,[6] Alexander Jarde,[7] Mustapha Bittaye,[8] Stanley Ulijaszek,[9] Caroline Fall,[2] Clive Osmond,[2] Andrew Prentice,[10] Fredrik Karpe[1,11]

► Prepublication history and supplemental material for this paper is available online. To view these files, please visit the journal online (http://dx.doi.org/10.1136/bmjopen-2020-042069).

For numbered affiliations see end of article.

**Correspondence to**
Professor Fredrik Karpe;
Fredrik.karpe@ocdem.ox.ac.uk

## ABSTRACT

**Introduction** Lifestyle modification is the mainstay of gestational diabetes mellitus (GDM) prevention. However, clinical trials evaluating the safety and efficacy of diet or physical activity (PA) in low-income and middle-income settings such as Africa and India are lacking. This trial aims to evaluate the efficacy of yoghurt consumption and increased PA (daily walking) in reducing GDM incidence in high-risk pregnant women.

**Methods and analysis** The study is a 2×2 factorial, open-labelled, multicentre randomised controlled trial to be conducted in Vellore, South India and The Gambia, West Africa. 'High-risk' pregnant women (n=1856) aged ≥18 years and ≤16 weeks of gestational age, with at least one risk factor for developing GDM, will be randomised to either (1) yoghurt (2) PA (3) yoghurt +PA or (4) standard antenatal care. Participants will be followed until 32 weeks of gestation with total active intervention lasting for a minimum of 16 weeks. The primary endpoint is GDM incidence at 26–28 weeks diagnosed using International Association of the Diabetes and Pregnancy Study Groups criteria or elevated fasting glucose (≥5.1 mmol/L) at 32 weeks. Secondary endpoints include absolute values of fasting plasma glucose concentration at 32 weeks gestation, maternal blood pressure, gestational weight gain, intrapartum and neonatal outcomes. Analysis will be both by intention to treat and per-protocol. Continuous outcome measurements will be analysed using multiple linear regression and binary variables by logistic regression.

**Ethics and dissemination** The study is approved by Oxford Tropical Research Ethics Committee (44–18), ethics committees of the Christian Medical College, Vellore (IRB 11367) and MRCG Scientific Coordinating Committee (SCC 1645) and The Gambia Government/MRCG joint ethics committee (L2020.E15). Findings of the study will be published in peer-reviewed scientific journals and presented in conferences.

**Trial registration number** ISRCTN18467720.

## Strengths and limitations of this study

► This study will be the largest lifestyle intervention trial among high-risk gestational diabetes mellitus (GDM) mothers in India and Africa to evaluate the efficacy of daily yoghurt consumption and physical activity in reducing GDM incidence.

► The factorial design allows to evaluate the effectiveness of independent and combined effects of yoghurt and physical activity on GDM incidence as well as test for interactions simultaneously.

► The inclusion of participants from two different low-income and middle-income settings with diverse cultural background, socioeconomic status, varied life-style behaviour and access to healthcare, will provide robust evidence of generalisability of these results across countries.

► Due to the nature of interventions, it is not possible to blind the study participants or research investigators, which may introduce bias and increase the risk of confounding.

► The selective screening and intervention in high-risk women using low-cost interventions is suitable for GDM prevention in low-resource settings such as India and Africa.

## BACKGROUND

Gestational diabetes mellitus (GDM) accounts for 86% of hyperglycaemia during the pregnancy[1] and is associated with significant intrapartum complications, perinatal morbidity and long-term risk of developing type 2 diabetes (T2D) and cardiovascular disease in both the mother and the offspring.[2–4] Consequent to the global rise in obesity prevalence and the introduction of more stringent diagnostic criteria by the International Association

of the Diabetes and Pregnancy Study Groups (IADPSG), the global prevalence of GDM has been estimated at 17.8% (range 9.3%–25.5%).[5] In low-income and middle-income countries (LMICs), especially in India and Africa, there are few nationwide estimates of GDM burden. Cross-sectional studies using variable diagnostic criteria report GDM prevalence ranging between 10%–16% in India[6 7] and 11%–14% in Africa,[8 9] which is higher than the reported prevalence from high-income countries like the UK (2%–3% using National Institute for Health and Care Excellence guidelines) and USA (2%–10% using various criteria including IADPSG).[10 11] Direct comparisons of these estimates may not be appropriate due to inconsistency in diagnostic criteria. Nevertheless, it seems clear that LMICs are facing a high burden of GDM.

In normal pregnancy, the physiological response to reduced insulin-mediated glucose removal is an increase in insulin secretion to maintain normoglycaemia.[12] In GDM, insufficient β-cell plasticity leads to an inability to secrete adequate quantities of insulin to counterbalance the insulin resistance (IR), and this process is accelerated in obesity. Women with chronic IR states such as prepregnancy excess body weight or obesity, impaired glucose tolerance prior to pregnancy, polycystic ovarian syndrome (PCOS), previous GDM, advanced maternal age and rapid gestational weight gain (GWG), have a higher risk for developing glucose intolerance during pregnancy.[13] The hyperglycaemia of GDM rapidly abates following delivery and normoglycaemia usually returns within 12 weeks. However, 50%–70% of GDM mothers progress to T2D within 5–10 years postpartum[14] suggesting that GDM may be a prodrome of 'common T2D' and may reflect the underlying T2D frequency in the community.[15]

### Screening for GDM

Following the demonstration of a linear relationship between maternal glycaemia and obstetric and perinatal outcomes by the Hyperglycaemic and Adverse Pregnancy Outcome study,[16 17] the IADPSG recommends universal screening of all women at first antenatal visit, without restriction to high-risk pregnant women, in order to identify all potential GDM cases.[18] However, some countries have adopted selective screening, despite the possibility of missing over 40% of GDM cases.[19] In low-economic setting like the LMICs, this approach may be more cost-effective and oral glucose tolerance test (OGTT) may be less beneficial to low risk women.[20]

### GDM management

The main goal of lifestyle modification using diet and physical activity (PA) in pregnancy is to prevent excessive GWG, improve maternofetal outcomes and prevent the onset/subsequent progression to GDM/T2D in both the mother and the new born.[21] In order to maximise the benefit of such interventions, the best strategy is likely to target high-risk populations by initiating early interventions during the 'critical window', that is, before insufficient β-cell plasticity sets in[22] (figure 1).

### Dietary interventions in pregnancy

Meta-analysis of dietary intervention trials in pregnant women has shown that diet modification in any form resulted in a 61% risk reduction in GDM incidence (relative risk (RR) 0.39, 95% CI 0.23 to 0.69).[23–26] Besides macronutrient modification, dairy products, particularly yoghurt has been shown to lower the risk of T2D,[27] through mechanisms such as reduction in inflammatory signalling and gut permeability giving rise to a range of putative signals from the gut to the systemic circulation which ameliorate metabolic endotoxaemia.[28–30] Differences in gut microbial composition between healthy pregnant and GDM women provide evidence of a potential role of altered gut microbiota in pregnancy-related metabolic dysfunction.[31 32]

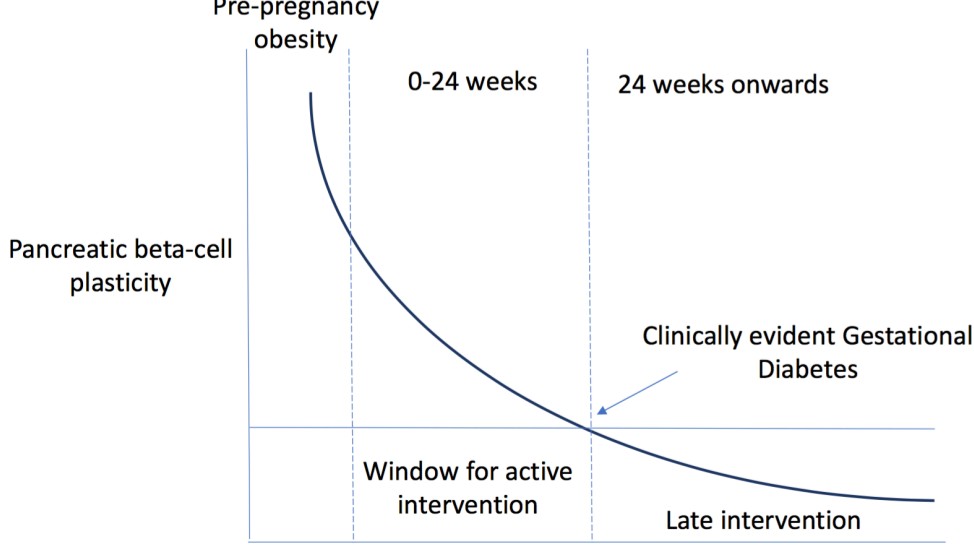

**Figure 1** Relationship between pancreatic beta cell plasticity with advancement in gestational age.

Although exact mechanisms of the positive effects of the manipulating the gut microbiome during pregnancy is not studied, evidence from epidemiological data suggest that using fermented dairy products or probiotics prior to and during the pregnancy has positive impact on maternal weight, blood pressure (BP), IR and plasma lipid profile.[31 33–37] The 'Probiotics and Pregnancy Outcome' study showed that incident GDM was lower in women supplemented with probiotics compared with diet/placebo or control groups (16% vs 36% vs 34%).[38] The Nutrition, Allergy, Mucosal immunology and Intestinal microbiota study showed reductions in glucose and insulin concentration, GDM incidence and central adiposity, possibly mediated by anti-inflammatory cytokine production in probiotic supplemented pregnant women.[39 40] Multiple pieces of evidence also support a positive relationship between probiotic yoghurt supplementation during pregnancy and improved glycaemic profile in women with GDM or obesity.[41–43]

## Physical activity in pregnancy

PA directly impacts non-insulin mediated glucose utilisation through translocation of glucose transporter type 4 on skeletal muscle.[44 45] PA during the pregnancy is safe and American College of Obstetrics and Gynaecology (ACOG) guidelines recommend ≥30 min of moderate PA performed at an intensity of 3–6 metabolic equivalents, which corresponds to brisk walking at 5–7 km/hour on most days of the week.[46]

Evidence of a beneficial effect from PA-intervention during the pregnancy mostly come from high-income countries and include a wide range of exercise programme that vary in their intensity, duration, frequency and assessment techniques. Results show that regular PA in any form improves maternal (hyperglycaemia, hypertension, weight gain and cardiovascular function),[47] obstetric (pre-eclampsia, shoulder dystocia, preterm births) and neonatal outcomes (low birthweight or small-for-gestational age (GA), decreased fat mass, improved stress tolerance and advanced neurobehavioural maturation).[48]

Meta-analysis of randomised controlled trials (RCTs) of PA intervention (eight studies, n=1441) in obese/overweight women has shown a 24% reduction in GDM incidence in the intervention groups (RR 0.76, 95% CI 0.56 to 1.03; p=0.07) with moderate heterogeneity between studies ($I^2$=50%, p=0.05).[49] Some studies have shown negative or neutral effects, including a Cochrane review (five studies, n=1115) of PA on GDM incidence,[50–52] and this is likely to be related to study heterogeneity, bias, type of exercise, timing, duration and intensity of exercise and also subjective assessments. Nevertheless, the potential beneficial effects of effective PA intervention in pregnancy should not be underestimated.

The knowledge gap on effectiveness of PA interventions in LMIC settings is particularly challenging in women from Asian Indian and African backgrounds where the baseline PA level is generally considered low. The ability to engage pregnant women from diverse ethnic groups will also be determined by population-specific socioeconomic and cultural factors.

## Combined approaches in pregnancy

Targeting multiple risk factors simultaneously may have a synergistic effect to reduce GDM. Combined approaches have shown favourable improvements in maternal and fetal outcomes, particularly GWG, which is a potentially modifiable risk factor for GDM development.[53–57] Two RCTs using combined interventions in high-risk women (body mass index (BMI) ≥30 kg/m$^2$), the UPBEAT (UK Pregnancies Better Eating and Activity Trial)[58] and RADIEL trial (Gestational diabetes mellitus can be prevented by lifestyle intervention: the Finnish gestational diabetes prevention study),[59] showed 0.55 kg and 0.58 kg reduction in GWG, respectively. The RADIEL trial additionally showed about 36% reduction in GDM incidence (crude RR 0.64; 95% CI 0.38 to 1.09) in the intervention group. A recent Cochrane review of combined interventions showed a tendency towards reduced GDM risk (RR 0.85, 95% CI 0.71 to 1.01; 6633 women, 19 RCTs) and reduced weight gain (mean difference –0.89 kg, 95% CI –1.39 to –0.40; 5052 women, 16 RCTs).[60]

## HYPOTHESIS

We hypothesise that daily yoghurt consumption and/or increased PA will reduce the risk of developing GDM in 'high-risk' pregnant women.

## AIMS

To examine if daily yoghurt consumption and/or PA beginning from ≤16 weeks of gestation in 'high-risk' pregnant women can reduce the incidence of GDM in two LMIC settings.

## OBJECTIVES
### Primary objective

To establish the efficacy and safety of low-cost interventions (fermented yoghurt consumption/daily walking/both) to prevent GDM in two LMIC settings.

### Secondary objectives

1. To evaluate the effect of yoghurt/daily walking/combination of both on fasting glucose concentration at 32 weeks of gestation, GWG, BP, intrapartum and neonatal outcomes.
2. To identify barriers that prevent participation/adherence to lifestyle changes during pregnancy and thus design culturally and socially acceptable, cost-effective interventions that can be implemented successfully in LMICs.

## CENTRES

Christian Medical College, Vellore, India.
MRC unit at LSHTM, Fajara, The Gambia.

## DESIGN

The study is a 2×2 factorial design and eligible women will be randomised to either one of the arms—(1) yoghurt (2) PA (3) yoghurt+PA and (4) standard care.

## ELIGIBILITY CRITERIA

The study will include pregnant women aged ≥18 years and GA ≤16 weeks and meeting at least one risk-factor for GDM (booking BMI ≥25 kg/m$^2$, age ≥25 years; a first-degree relative with diabetes; a previous pregnancy complicated by GDM, pre-eclampsia/eclampsia and/or a large baby (≥3.5 kg); a history of PCOS/impaired fasting glucose) and not currently on any medication that is known to interfere with glucose metabolism.

### Exclusion criteria

▶ GDM diagnosed prior to screening visit based on IADPSG criteria or documented raised glycosylated haemoglobin (HbA1c), that is, either fasting glucose ≥5.1 mmol/L or 1 hour glucose≥10.0 mmol/L or 2 hour glucose ≥8.5 mmol/L, or a documented HbA1c of ≥6.5% at first booking.

▶ History of pregestational diabetes, severe hyperemesis in the first trimester.

▶ History of recurrent (≥2) first trimester spontaneous abortions, stillbirth, significant antepartum or postpartum haemorrhage in previous pregnancies.

▶ Multiple gestation in the current pregnancy.

▶ Uncontrolled pregestational or gestational hypertension (BP >150/100 mm Hg) on treatment.

▶ Previous child born with congenital anomalies.

▶ Pregnancy following in vitro fertilisation or any assisted reproductive technology.

▶ Previous or current psychiatric illness on medication, epileptic seizures or on antiepileptic medication,

▶ Women meeting absolute contraindications for PA during the pregnancy as recommended by the ACOG (heart disease, restrictive lung disease, incompetent cervix/cerclage, pregnancies at risk for premature labour, gestational hypertension, severe anaemia).

## METHODS

Women attending obstetric outpatient clinics of the Christian Medical College, Vellore, India and MRC Unit in The Gambia operating in four centres in The Gambia (Brikama Major Health Centre, Bundung Maternal and Child Hospital, Barfrow Medical Centre and Edward Francis Small Teaching Hospital) will be informed about the study and assessed for eligibility. Following informed consent, women enrolled in the study will be asked to attend five scheduled visits (screening, run-in phase, randomisation, visits 1, 2 and 3) as shown in figure 2 and detailed in the protocol (online supplemental file 1). The study is expected to run from September 2018 to August 2021.

## RANDOMISATION

Participants will be randomly allocated to one of the following arms (1) yoghurt (2) PA (3) yoghurt +PA or (4)

standard care following an allocation schedule prestratified for centre, age and BMI. A variable block randomisation, stratified by centre and then by age, (<25 and ≥25 years) and BMI (<25 and ≥25 kg/m$^2$) will be generated, using a password-protected access database developed by an independent statistician not involved with trial participants. If a woman has more than one risk factor, she will be allocated to the strata depending on the first of the risk factors in the hierarchical block randomisation procedure (age or BMI). In India, the allocation will be performed using a computer interface (Access database), while separate computer-generated randomisation lists for each strata combination (age X BMI) will be prepared for each health facility in the Gambia. Due to logistic limitations, this will be implemented using pre-issued opaque, sealed envelopes containing the allocation code.

## INTERVENTIONS
### Dietary intervention

Yoghurt will be prepared locally at the dietary department in CMC, Vellore and at Kombo Dairy farm in The Gambia based on a common recipe and using the same starter culture containing *Lactobacillus* spp and *Bifidobacterium* spp at 10$^7$ CFU. Yoghurt will be prepared in batches and an internal quality check will be performed to ensure that the bacterial species and colony count are maintained with each batch. Participants in the active yoghurt arm will consume 200 g/day of regular, unflavoured, fermented yoghurt throughout the active intervention period independent of the timing of their regular meal. Yoghurt will be dispensed either daily or weekly as feasible by the study centre and possibility of storage in refrigerator at participants' homes when not in use. Compliance with yoghurt will be assessed using pot count by field workers. All participants will receive dietary counselling, will not be advised to stop their regular dairy consumption, assuming that any effect on the outcome will be related to the trial yoghurt consumed. Information on background dairy consumption will be collected at each scheduled visit using a standardised DAIRY consumption questionnaire.

### PA intervention

The PA activity intervention will include daily walking to increase the individual target step count, monitored using a wrist-mounted accelerometer device (Garmin Vivofit 4 fitness band).

The baseline step count will be recorded in all participants (blinding the step count on the device) for 7 days, with no emphasis on step count and/or activity during the run-in phase. At randomisation, baseline step count will be calculated as the average over 7 days including 2 weekend days. An individual target will be set, to a 40% or greater increase in daily step count compared with the baseline reading. The increment of 40% was based on a feasibility estimation. Systematic reviews have shown that simple pedometer interventions can lead to a sustained increase of around 2000–2500 steps a day in a great number of populations and settings.[61 62] The background step count in these populations

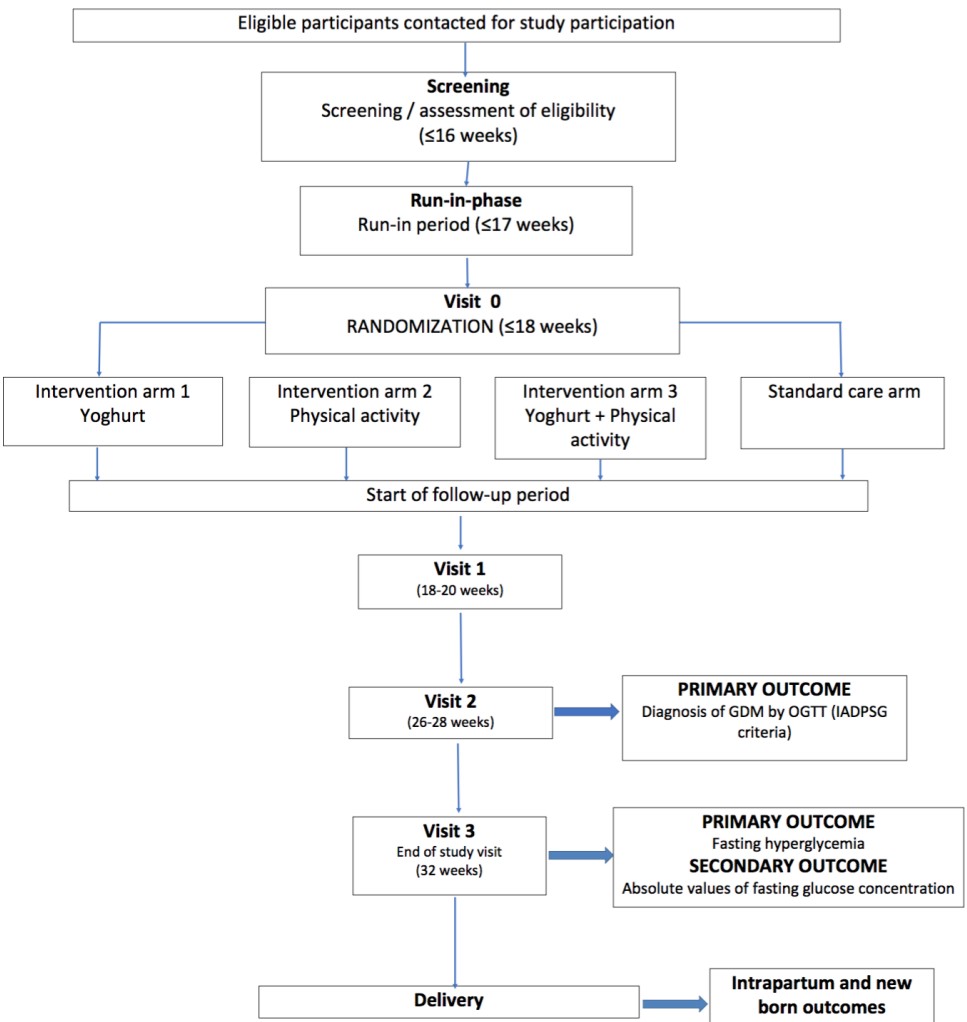

**Figure 2** Flow chart of the study recruitment and scheduled visits. GDM, gestational diabetes mellitus; IADPSG, International Association of the Diabetes and Pregnancy Study Groups; OGTT, oral glucose tolerance test.

was reasonably low (~5000), which will mean that 2000 additional step represent 40%.

The step count reading on the device will be unblinded after randomisation for women in the active PA arms, to provide visual feed back to the participants and to motivate them to achieve their daily target step count. Women in the non-PA arm will continue to use their fitness bands blinded throughout the study to measure their routine PA behaviour. Step count data will be periodically uploaded using Garmin application by the field worker to quantify compliance monitoring and counselling to participants in the PA arm. As the device is water resistant, the participants would be instructed to wear it day and night.

## STUDY ASSESSMENT
### Clinical measurements
Detailed study-related procedures, as outlined in table 1 and routine antenatal examination will be carried out at all study visits. All measurements will be standardised between the two study centres and will be carried out as detailed in study-specific standard operating procedures (SOPs).

### Laboratory tests
Study participants will undergo a 75 g OGTT at screening and visit 2 (week 26–28 GA) following an 8-hour overnight fast. Additional fasting plasma glucose will be measured at the end-of-study visit (week 32). Blood glucose measurements will be analysed using a Roche Cobas 800 autoanalyser at respective laboratories (Department of Clinical Biochemistry in Vellore, India and the MRCG at LSHTM, The Gambia). Other routine antenatal laboratory tests, including haemoglobin will be done as a part of routine antenatal care according to local guidelines.

### GDM ascertainment
IADPSG criteria will be used to diagnose GDM using an OGTT at weeks 26–28. GDM diagnosis will be established if either fasting glucose ≥5.1 mmol/L or 1-hour glucose ≥10.0 mmol/L or 2-hour glucose ≥8.5 mmol/L.

### Questionnaire-based assessments
Standardised questionnaires will be used to obtain information on social demographics, background dairy consumption and to identify barriers to lifestyle intervention

**Table 1** Study-related procedures at each study visit

| Procedures | Visit −2 Screening ≤16 weeks | Visit −1 Run-in-phase ≤17 weeks | Visit 0 Randomisation Week ≤18 | Visit 1 Week 18–20 | Visit 2 Week 26–28 | Visit 3 Week 32 | Delivery |
|---|---|---|---|---|---|---|---|
| Informed consent | X | | | | | | |
| Eligibility assessment | X | | | | | | |
| Demographics | X | | | | | | |
| History | X | | | | | | |
| Height | X | | | | | | |
| Weight | X | X | X | X | X | X | X |
| Blood pressure | X | X | X | X | X | X | X |
| Antenatal examination | X | X | X | X | X | | |
| Dairy consumption questionnaire | | X | X | X | X | X | |
| Barriers questionnaire | X | | | | | X | |
| Baseline PA monitoring with Vivofit 4 | | X | | | | | |
| PA step-count monitoring | | | X | X | X | X | |
| Randomisation | | | X | | | | |
| OGTT | X | | | | X | | |
| Dating scan | X | | | | | | |
| Morphology scan | | | | X | | | |
| Ultrasound scan (growth) | | | | | | X | |
| Fasting plasma glucose | | | | | X | | |
| Intrapartum assessment (mother) | | | | | | | X |
| New born assessment | | | | | | | X |

OGTT, oral glucose tolerance test; PA, physical activity.

during pregnancy in all participating women. The barriers questionnaire is specifically designed to address ethnic-specific social and cultural barriers to lifestyle intervention during the pregnancy in both populations and would be administered at the start and the end of the study. The DAIRY consumption questionnaire used in this study is a modification the Dutch Dairy Food Frequency Questionnaire (Wageningen University, Department Human Nutrition, 2005).[63] The questions relate to consumption of various dairy products divided into six different groups: total dairy, cheese, milk, nonfermented milk products, fermented milk and other dairy products. Flavoured milk, puddings and cream are all part of the non-fermented milk product group. The fermented milk products consist of yoghurt, yoghurt drinks and butter milk. Additional modifications to include other dairy preparations that are locally consumed in both countries are included.

## Ultrasound scans

Ultrasound scans will be performed by an ultrasound-trained obstetrician on Voluson GE S-8 (in India) and Aloka USI-145 (in Gambia) ultrasound machines on all participates at three visits, (1) at screening (dating) (2) morphology scan at visit 1 (18–20 weeks GA) and (3) at visit 3 (week 32) for growth monitoring. Both centres will follow standard protocols as described in study SOPs. Evidence of fetal congenital anomaly, placental malformation or malposition or abnormal ultrasound features that can compromise pregnancy outcome will lead to termination of the participant's involvement in the study and referral for the normal care pathway at each participating centre.

## OUTCOMES

### Primary outcome

Incidence of GDM diagnosed using IADPSG criteria between 26 and 28 weeks of gestation or if the fasting plasma glucose concentration is ≥5.1 mmol/L at 32 weeks of gestation.

### Secondary outcomes

Effect of the intervention on fasting blood glucose concentration at 32 weeks, GWG and BP from randomisation to week 32, proportion of women undergoing instrumental/

caesarean delivery, proportion of women developing pre-eclampsia and eclampsia, blood loss during delivery (subjective assessment by the obstetrician), postpartum haemorrhage, preterm births (<37 weeks GA), fetal macrosomia defined as birth weight >2 SDs above the population-specific mean in each setting, birth weight and length, APGAR score at 1 and 5 min and responses to questionnaire based assessments of barriers to lifestyle interventions in pregnancy.

## ADVERSE/SERIOUS AES AND REPORTING

All adverse event (AE) and serious AEs (SAEs) will be captured in respective electronic Case Report Forms (eCRFs). Details of all events including onset date, relationship to the study intervention, treatment given and stop date will be captured in the respective AE report form. The PI will report all SAEs without filtration, whether or not related to the intervention, within 24 hours of becoming aware of the event, to the sponsor. All SAEs will be notified to the respective Ethics Committee within seven calendar days if fatal or life-threatening, and all others within 15 calendar days. All AEs and SAEs will also be notified to the Data Safety and Monitoring Committee (DSMC). Expected SAEs which are protocol-defined exceptions to SAE reporting will be captured and reported to respective ethics committees (ECs) and DSMC as outlined in the protocol (online supplemental file 1).

## STATISTICAL ANALYSIS PLAN INCLUDING SAMPLE SIZE AND POWER CALCULATION

### Sample size

The sample size calculation was performed by simulation (in R V.3.5.2) using an algorithm as described in detail in the protocol (online supplemental file 1). After 10 000 repetitions, we estimated the sample size to be 1856 high-risk women (928 from each centre). The sample size is estimated to have 80% power to detect a 33% reduction in the incidence of GDM, for the main effects of diet and PA, with a family-wise significance level of 5% for a two-sided test (ie, each effect was tested at a significance level of 2.5%). It was assumed that participants could be pooled across centres (no interaction between interventions and country at the 5% level). The sample size was inflated to allow for a 15% drop-out rate in each of the four strata, with drop outs defined as non-compliant participants (ie, no risk reduction in their incidence of GDM).

### Data analysis

Data will be analysed using both (1) the intention-to-treat principle, including all enrolled participants regardless of their adherence to the proposed intervention and (2) a per protocol approach, including only those participants adhering to at least 75% of the proposed intervention. In addition, we will run a sensitivity analysis to assess the effect of protocol deviation compared with 100% adherence.

### Descriptive analyses at baseline

We will describe baseline characteristics of the four groups (Control, yoghurt, PA, and yoghurt +PA) using means and standard deviations (normally distributed quantitative variables), medians and interquartile ranges (non-normally distributed quantitative variables) or proportions (categorical variables).

### Primary analysis

For our primary outcome (presence or absence of GDM), we will use multiple logistic regression models and results presented as OR and 95% CI. We will examine the assumption that there are no interaction effects between study centre and each of the interventions by fitting a multiple logistic regression model.

We will build two covariate adjusted models. First, we will assess the relative difference between centres in the effects of each intervention on GDM by adding a term for study centre. Second, we will fit a covariate (age, prepregnancy BMI, previous history of GDM and GA) adjusted model adding further terms for each, as well as relevant interaction terms, to remove bias from the estimate of the treatment effect on the primary outcome. Each covariate will be added to the unadjusted model separately first and then added one at a time and kept in the model if their significance level is <10%.

### Secondary analyses

Continuous variables will be analysed using multiple linear regression and binary variables by multiple logistic regression. For each secondary outcome, we will fit a simple model with only terms for each intervention, and a covariate adjusted model including terms for study centre and the remaining pre-specified covariates and relevant interactions.

## DATA COLLECTION

Research data will be collected using eCRFs on tablets by designated study staff after training. Where there is internet connectivity, collected data will be stored directly onto the central server housed at MRC, Fajara and where there is no/unreliable connectivity, data will be stored locally on the tablet and later synchronised to the central servers. The accelerometer data used for PA monitoring will be directly synced with the Garmin app on the tablets on a weekly basis and transferred to eCRFs. To ensure standardisation of processes, SOPs with respect to trial management, quality assurance, data management, information technology (IT) and security and statistics will be adhered to.

## COMMITTEE OVERSIGHTS

The study will be conducted in accordance with the approved protocol, Good Clinical Practice regulations

and SOPs. The implementation of the study will be overseen by the trial steering committee (TSC). An independent DSMC, composed of experts in clinical trials/epidemiology, biostatistics and obstetrics will meet 6-monthly to review the progress of the trial (recruitment, compliance, lost to follow-up), clinical study safety data, data quality and assess and/or adjudicate all events of deaths, the GDM event rate and key secondary/safety outcomes and provide feedback to the TSC. The DSMC will also meet if required, more frequently depending on the enrolment rate, safety issues and AEs. A copy of the full study protocol and all amendments would be submitted to DSMC prior to study initiation and subsequently. Thereafter, periodic reports including summary of enrolment, demographics, protocol compliance and safety will be compiled by the study statistician and submitted at least 1 week prior to the scheduled meeting. The board will advise on continuation or stopping the trial based on safety and efficacy.

## ETHICS AND DISSEMINATION
### Ethical issues

The proposed interventions are considered generally safe both for maternal and fetal health. Participation in this trial is voluntary and subject to informed consent. A sample informed consent statement is available in the study protocol (online supplemental file 1). Participants who develop GDM at the visit 2 OGTT will be started on pharmacological therapy (as per local management guidelines) and results will be analysed as per intention to treat. Research participants can withdraw from the study at any time during the trial. The Standard Protocol Items: Recommendations for Interventional Trials Checklist for the protocol is detailed in online supplemental file 2.

### Dissemination

Results of the study will be presented at major national and international conferences, and publish in peer-reviewed journals, ensuring open access and the highest possible impact. The deidentified data will be openly available to the scientific community after publication of initial results. We will interact with national and international health agencies and governments to stimulate translation of our findings into policy and expect our research to influence policy for mothers and children within the next 5–10 years.

### Patient and public involvement

The trial was informed by patients' need for effective preventive therapies for GDM since randomised trial data are lacking in these populations. Patients were not involved in the design, conduct or reporting of the study. A summary of the main trial results will be disseminated to the general public and study participants after the completion of the study.

## DISCUSSION

GDM poses increased risks to maternal and child health. LMICs face numerous challenges to GDM care at various levels. These include (1) late second trimester screening for GDM, (2) lack of standardised nationwide guidelines for diagnosis, (3) interventions targeting only women who are diagnosed with GDM around 24–28 weeks, (4) lack of knowledge and awareness of GDM and its health implications and (5) sociocultural barriers to early antenatal care and life-style interventions during pregnancy. It seems very likely that early screening and establishment of positive lifestyle changes beginning from conception, or even prior to conception, particularly in selected high-risk women and especially in low-resource settings, will reduce the burden of GDM as well as improve both immediate obstetric outcomes and long-term health of the mother and the new born. Primary prevention of GDM, rather than treatment initiated after GDM onset is likely to lead to both economic and individual health benefits.

The PRIMORDIAL study is timely given the increasing prevalence of both diabetes and obesity in countries like Africa and India. Our study provides a unique opportunity to test simple life-style interventions during pregnancy in the LMIC settings, which have never been undertaken. The results will add to the existing evidence of beneficial effects of yoghurt on several pregnancy-related outcomes.

The study is expected to bridge a gap in knowledge of the potential effectiveness of GDM prevention in two LMIC settings and create opportunities for capacity building in both India and The Gambia. We believe the trial can provide information that will aid in policy guidelines relating to GDM screening and management and also suggest areas that may require further research.

**Author affiliations**
[1]Oxford Centre for Diabetes, Endocrinology and Metabolism, University of Oxford, Oxford, UK
[2]MRC Lifecourse Epidemiology Unit, University of Southampton, Southampton, UK
[3]MRC Nutrition Group, MRC Unit The Gambia at LSHTM, Banjul, The Gambia
[4]Obstetrics and Gynaecology, Christian Medical College and Hospital Vellore, Vellore, Tamil Nadu, India
[5]Data Management, MRC Unit The Gambia at LSHTM, Banjul, The Gambia
[6]Department for Health, University of Bath, Bath, UK
[7]Statistics, MRC Unit The Gambia at LSHTM, Banjul, The Gambia
[8]Obstetrics and Gynaecology, MRC Unit The Gambia at LSHTM, Banjul, The Gambia
[9]School of Anthropology, University of Oxford, Oxford, UK
[10]MRC International Nutrition Group, London School of Hygiene & Tropical Medicine, London, UK
[11]NIHR Oxford Biomedical Centre, Oxford University Hospital Trust and University of Oxford, Oxford, UK

**Contributors** SKV and FK conceived and designed the study. SKV, DT, CF, CO, SU, AP and FK drafted the original grant proposal and trial protocol. CO, FC and AJ provided methodological and data analysis plan. JM, SR, MJ, MB and AP provided expertise in the pregnancy clinical outcomes. DT, SU, CF and AP provided expertise in study interventions. FC and AJ will contribute significantly to the data management and central data monitoring. SKV drafted the original protocol. SKV, MJ, OJ, MB, JM and SR have responsibilities for day-to-day running of the trial including participant recruitment, data collection and liaising with other sites. SKV and FK drafted the first version of this manuscript. All authors critically reviewed and approved the final version of the manuscript. All authors agreed to be accountable for all aspects of the work.

**Funding** This study is supported by grants jointly from Medical Research Council, UK (MR/R020345/1) and Newton Fund- Director of Biotechnology, India (BT/IN/DBT-MRC/DIFD/JM/12/2018–19) The study is partly supported by the Global Challenges Research fund, University of Oxford, UK (0006138)

**Disclaimer** The views expressed in this publication are those of the author(s) and not necessarily those of the MRC, UK, GCRF, UK, DBT, India, healthcare systems or competent authorities.

**Competing interests** None declared.

**Patient consent for publication** Not required.

**Provenance and peer review** Not commissioned; externally peer reviewed.

**ORCID iD**
Senthil K Vasan http://orcid.org/0000-0002-3630-6568

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
