## [Reviewer comments · BMJ Open]

ARTICLE DETAILS

TITLE (PROVISIONAL)	Protocol for a randomised controlled trial: Pregnancy Related Interventions in Mothers at risk for gestational Diabetes in Asian India and Low and middle income countries (PRIMORDIAL Study)
AUTHORS	Vasan, Senthil; Jobe, Modou; Mathews, Jiji; Cole, Fatoumata; Rathore, Swathi; Jarjou, Ousman; Thompson, Dylan; Jarde, Alexander; Bittaye, Mustapha; Ulijaszek, Stanley; Fall, Caroline; Osmond, Clive; Prentice, Andrew; Karpe, Fredrik

VERSION 1 – REVIEW

REVIEWER	Joao Guilherme Alves Instituto de Medicina Integral Prof. Fernando Figueira (IMIP) Brazil
REVIEW RETURNED	11-Aug-2020

GENERAL COMMENTS	This study presents a protocol for a randomized controlled trial, Pregnancy Related Interventions in Mothers at risk for gestational Diabetes in Asian India and Low and middle income countries (PRIMORDIAL Study). This is a well-designed trial. I only have a few comments. 1 – Title: "... in Asian India and Low and middle income countries". The mention of Africa is missing. Do both regions are low and middle-income countries? 2 – Background: "... particularly yoghurt has been shown to lower the risk of T2D, 27 probably because of their probiotic effect of reducing inflammatory signaling and gut permeability giving rise to a range of putative signals from the gut to the systemic circulation.²⁸⁻³⁰ However, it should be noted that these studies were not in pregnant women". However, further ahead authors show studies on the effect of yogurt on plasma glucose/insulin resistance in pregnant women. 3 – This reference is missing: Chen X et al. Association between Probiotic Yogurt Intake and Gestational Diabetes Mellitus: A Case-Control Study. Iran J Public Health. 2019;48(7):1248-1256. 4 - How randomization will be performed in participants with more than one risk factor for GDM? Example: BMI \geq25kg/m², age \geq25 years, and a first-degree relative with diabetes or a previous pregnancy complicated by GDM, pre-eclampsia/eclampsia or a large baby (\geq3.5kg) or a history of PCOD/impaired fasting glucose. 5 – How obese pregnant women (BMI>30Kg/m²) will be handled? And mothers > 40 years-old? 6 – It seems that physical activity patterns before pregnancy may also influence some gestational outcomes. Do participants have a previous evaluation of the physical activity pattern? I suggest IPAQ or PPAQ 7 – Intolerance to lactose should be an exclusion criterion.
--

	8 - Step count data will be periodically uploaded using Garmin application to provide visual feedback. How participants will be trained to do this? Are all participants literate? 9 – What does mean “a qualified obstetrician” to perform ultrasound scans? 10 – How blood loss during delivery will be measured? What is the concept of post-partum haemorrhage? 11 – Physical activity decreases during pregnancy. It will be considered an increase of 40% in daily steps during all the weeks of study. Why 40%? Does this parameter will used during all periods of follow-up (16 weeks)? 12 – Heart rate will not be measured during walking. But it should be. How will the walk be guided? Brisk-walking (5-7Km/h)? Mild or moderate intensity? 13 – Why gestational weight gain will be calculated from admission and not before pregnancy? 13 – Does a standard questionnaire will be used to obtain information on dairy consumption during the intervention (16 weeks)? 14 – Garmin vivofit 4 fitness band is an accelerometer or a pedometer? Does it measure only step counts? It will be used all day including sleep and shower time? 15 – Discussion could approach benefits and facilities for implementing both of the interventions.
--	---

REVIEWER	Luat Cong Nguyen National Institute of Hygiene and Epidemiology Vietnam
REVIEW RETURNED	29-Oct-2020

GENERAL COMMENTS	The protocol is well-prepared. The success of this study will add evidence for the prevention of GDM. The authors should describe the study strengths and limitations.
--

REVIEWER	Polina Popova
REVIEW RETURNED	21-Dec-2020

GENERAL COMMENTS	The study addresses an interesting question to evaluate the efficacy of fermented product consumption and physical activity increase or their combination in reducing GDM risk in a high-risk population. The protocol is well designed and well written. There is a small mistake in Line 7: «PCOD» instead of PCOS?
---

VERSION 1 – AUTHOR RESPONSE

Reviewer: 1

Dr. João Alves, Instituto de Medicina Integral Prof. Fernando Figueira (IMIP)

Comments to the Author:

This study presents a protocol for a randomized controlled trial, Pregnancy Related Interventions in Mothers at risk for gestational Diabetes in Asian India and Low and middle income countries (PRIMORDIAL Study). This is a well-designed trial. I only have a few comments.

1 – Title: “... in Asian India and Low and middle income countries”. The mention of Africa is missing. Do both regions are low and middle-income countries?

Response: We agree with the reviewer that both India and Africa are low-and middle-income

countries and the title is not ideal. It would be better to remove Asian India and but if we do that the title of the paper is not the same as the funded trial, so we suggest no change. The title was based on the grant call (MRC-Newton Fund-DBT) which specifically aimed at tri-partnership between UK-India and any other LMIC. The main funding resources for UK and LMIC are from MRC, UK and for India by the DBT.

2 – Background: "... particularly yoghurt has been shown to lower the risk of T2D, 27 probably because of their probiotic effect of reducing inflammatory signaling and gut permeability giving rise to a range of putative signals from the gut to the systemic circulation.²⁸⁻³⁰ However, it should be noted that these studies were not in pregnant women". However, further ahead authors show studies on the effect of yogurt on plasma glucose/insulin resistance in pregnant women.

Response: Sorry for the lack of clarity here. The first part refers to mechanistic studies (Ref 28-30) in individuals with T2D. Although alterations in gut microbiome during pregnancy has been extensively reported in literature, exact mechanisms by which such alterations impact metabolic outcomes have not been studied. The effect of yoghurt on plasma glucose/insulin resistance (Ref 31, 33-36) are based on observational data from epidemiological studies.

We have now re-worded this statement as below to add more clarity

Besides macronutrient modification, dairy products, particularly yoghurt has been shown to lower the risk of T2D, 27 through mechanisms such as reduction in inflammatory signalling and gut permeability giving rise to a range of putative signals from the gut to the systemic circulation which ameliorate metabolic endotoxemia.²⁸⁻³⁰

Although exact mechanisms of the positive effects of the manipulating the gut microbiome during pregnancy is not studied, evidence from epidemiological data suggest that using fermented dairy products or probiotics during pregnancy has positive impact on maternal weight, blood pressure, insulin resistance and plasma lipid profile.^{31 33-36}

3 – This reference is missing: Chen X et al. Association between Probiotic Yogurt Intake and Gestational Diabetes Mellitus: A Case-Control Study. *Iran J Public Health*. 2019;48(7):1248-1256.

Response: We thank the reviewer for this reference to a case-control study in Chinese pregnant women, where the results show that self-reported probiotic yoghurt consumption during pregnancy reduced the risk of gestational diabetes by 29.2%. We have now included this study as reference 43 in the edited manuscript.

We have now included this new reference in the revised manuscript

4 - How randomization will be performed in participants with more than one risk factor for GDM?

Example: BMI ≥ 25 kg/m², age ≥ 25 years, and a first-degree relative with diabetes or a previous pregnancy complicated by GDM, pre-eclampsia/eclampsia or a large baby (≥ 3.5 kg) or a history of PCOD/impaired fasting glucose.

Response: We are not specifically separating out women with more than one risk factor. Instead we have a hierarchical block randomisation procedure based on that age and BMI have a substantial impact on the development of gestational diabetes, poor perinatal outcomes and increased risk of subsequent development of T2D compared with other risk factors. Therefore, we considered these as most important risk factors at randomisation. The randomisation is stratified by centre (India or The Gambia), and then by age, (<25 and ≥ 25 years) and BMI (<25 and ≥ 25 kg/m²). This implies that the risk factors (age and BMI) would be evenly distributed to all four groups. We rely on the randomization procedure to distribute other risk factors more or less evenly, as adding further stratification in the block randomisation was impractical/not feasible.

To address this, we have now added a line under section randomisation as below

“If a women has more than one risk factor, she will be allocated to the strata depending on the first of the risk factors she has in the hierarchical block randomisation procedure (age or BMI).”

5 – How obese pregnant women (BMI>30Kg/m²) will be handled? And mothers > 40 years-old?

Response: They will be included unless they fail one of the exclusion criteria. We believe it is clear enough in the paper that there is no upper limit for BMI or age.

6 – It seems that physical activity patterns before pregnancy may also influence some gestational outcomes. Do participants have a previous evaluation of the physical activity pattern? I suggest IPAQ or PPAQ

Response: We thank the reviewer for highlighting the importance of physical activity prior to enrolment in the study. Although GPAQ or IPAQ are good assessment tools for physical activity, they are subjective and are prone to recall bias. We believe that our objective measurement of baseline physical activity using accelerometers during the run-in-period is a very robust tool of baseline physical activity pattern.

7 – Intolerance to lactose should be an exclusion criterion.

Response: We fully agree with the reviewer and would include ‘intolerance to lactose’ as an exclusion criterion in the next amendment of the protocol. However, we prefer not to add this exclusion criterion in the current manuscript, as this would be a protocol amendment and would require ethical approval, before publication.

8 - Step count data will be periodically uploaded using Garmin application to provide visual feedback. How participants will be trained to do this? Are all participants literate?

Response: Step counts will be periodically uploaded either during the study visits/weekly group physical activity programme/ during home visits by the field worker. This would be done by the trained study staff, and not by the participants. Providing a visual feedback to the participants serve as a deliberate behaviour change strategy around reflection and achieving the prescribed targets. We have now made a change to the last sentence in this paragraph for clarification.

Old version: Step count data will be periodically uploaded using Garmin application to provide visual feedback, compliance monitoring and counselling to participants in the PA arm.

Revised in the manuscript: Step count data will be periodically uploaded using Garmin application by the field worker to quantify compliance monitoring and counselling to participants in the PA arm.

9 – What does mean “a qualified obstetrician” to perform ultrasound scans?

Response: A “qualified” obstetrician is one who has been trained to perform ultrasound scans. We realised that unlike in the west, not all obstetricians in the LMICs (especially Africa) are trained in obstetric ultrasounds, due to limited resources. Therefore, in this study scans would be performed by obstetricians who are qualified and trained in ultrasound scanning. We have edited this to ‘ultrasound-trained obstetrician’ and the word ‘qualified’ is removed.

10 – How blood loss during delivery will be measured? What is the concept of post-partum haemorrhage?

Response: Women with GDM are at higher risk of adverse maternal outcomes including increased blood loss during delivery and post-partum haemorrhage. In this study, “blood loss during delivery’

would be a subjective assessment by the obstetrician conducting the delivery. We will not measure blood loss, instead this will be a qualitative assessment at delivery and only made positive if the blood loss was clinically significant.

11 – Physical activity decreases during pregnancy. It will be considered an increase of 40% in daily steps during all the weeks of study. Why 40%? Does this parameter will used during all periods of follow-up (16 weeks)?

Response: We agree with the reviewer that physical activity decreases as pregnancy progresses. Our physical activity intervention is 40% increase from the baseline step count (assessed as average of 5-7day step count during the run-in-period). Participants will be encouraged to adhere to the fixed 40% increase (target step count) throughout the study. The 40% is based on previous pilot experiments conducted by our collaborator who found that a technology-enabled PA intervention successfully increased physical activity by >3,000 steps/day at 12 weeks which represented an increase in objectively assessed energy expenditure of almost 300 kCal per day (Prof Dylan Thompson, unpublished). More broadly, systematic reviews have shown that simple pedometer interventions can lead to a sustained increase of around 2000-2500 steps a day in a great number of populations and settings (e.g., Mansi et al 2014 BMC Musculoskeletal Dis 15:231; Bravata et al 2007 JAMA 298:2296-2304). Thus, based on our research and that of others we have a great deal of confidence that participants will be able to increase and sustain the 40% increase over the course of the study.

A statement to this is added in the revised manuscript as below:

An individual target will be set, to a 40% or greater increase in daily step count compared with the baseline reading. Systematic reviews have shown that simple pedometer interventions can lead to a sustained increase of around 2000-2500 steps a day in a great number of populations and settings (e.g., Mansi et al 2014 BMC Musculoskeletal Dis 15:231; Bravata et al 2007 JAMA 298:2296-2304). The background step count in these populations was reasonably low (~5000), which will mean that 2000 additional step represent 40%.

12 – Heart rate will not be measured during walking. But it should be. How will the walk be guided? Brisk-walking (5-7Km/h)? Mild or moderate intensity?

Response: Garmin Vivofit device is an activity tracker and this model does not record heart rate. We do not intend to use heart rate to guide the walking intervention. Our aim is to propose a realistic physical activity intervention which is achievable by pregnant mothers in the low- and middle-income settings where women are generally not physically active during pregnancy. This could be simply achieved by increasing their step count from baseline and not emphasising on intensity. Several studies have shown that a modest increase in steps per day is more important and associated with improved metabolic outcomes than intensity (Lee et al, JAMA int. Med 2019; Dwyer et al BMJ 2011) and recommendations for intensity, particularly during pregnancy may not be sustainable and lead to poor compliance and even drop-outs in the study.

13 – Why gestational weight gain will be calculated from admission and not before pregnancy?

Response: The objective of the study is not to evaluate the impact of pre-pregnancy weight on the outcomes, but to evaluate the effect of interventions on weight gain during pregnancy, therefore, pre-pregnancy weight gain is not relevant to our study. Gestational weight gain will be calculated from the time of first weight measurement at randomisation (when actual intervention begins) to the end of study visit. Data on pre-pregnancy weight could only be a subjective recall and prone to errors, and in such settings women may not even have a record of pre-pregnancy weight.

14 – Does a standard questionnaire will be used to obtain information on dairy consumption during the intervention (16 weeks)?

Response: Yes, we will be using a standard diary consumption questionnaire, which is a modification of the Dutch Dairy Food Frequency Questionnaire (Wageningen University, Department Human Nutrition, 2005). The questions relate to consumption of various dairy products divided into 6 different groups: total dairy, cheese, milk, nonfermented milk products, fermented milk, and other dairy products. Flavoured milk, puddings, and cream are all part of the nonfermented milk product group. The fermented milk products consist of yogurt, yogurt drinks, and butter milk. Additional modifications to include other dairy preparations that are locally consumed in both countries are included. We have included details of the DSAIRY consumption questionnaire in the revised manuscript.

15 – Garmin vivofit 4 fitness band is an accelerometer or a pedometer? Does it measure only step counts? It will be used all day including sleep and shower time?

Response: Garmin vivofit 4 fitness band is an accelerometer which records step counts and is water-resistant. According to the study protocol, participants are advised not to remove the fitness band from their wrists and wear it all day including sleep and shower time.

16 – Discussion could approach benefits and facilities for implementing both of the interventions.

Response: The facilities for implementing both of the interventions are already described in detail under the intervention sections. We have included the benefits of implementing such interventions in the revised manuscript under discussion section as below

We believe that our study is timely given the increasing prevalence of both diabetes and obesity in countries like Africa and India. Our study provides a unique opportunity to test simple life-style interventions during pregnancy in the LMIC settings, which have never been undertaken. The results will add to the existing evidence of beneficial effects of yoghurt on several pregnancy related outcomes. We believe that our results will inform policy change regarding universal screening for GDM, which will reduce the adverse maternal and new-born outcomes and in turn reduce the burden of diabetes in these countries.

Reviewer: 2

Mr. Cong Luat Nguyen

Comments to the Author:

The protocol is well-prepared. The success of this study will add evidence for the prevention of GDM. The authors should describe the study strengths and limitations.

Response: We thank the reviewer for their positive comment.

Reviewer: 3

Dr. Polina Popova, Nacional'nyj medicinskij issledovatel'skij centr imeni V A Almazova

Comments to the Author:

The study addresses an interesting question to evaluate the efficacy of fermented product consumption and physical activity increase or their combination in reducing GDM risk in a high-risk population. The protocol is well designed and well written.

There is a small mistake in Line 7: «PCOD» instead of PCOS?

Response: We thank the reviewer for their positive comment. As suggested, we have now changed PCOD to PCOS.

VERSION 2 – REVIEW

REVIEWER	Joao Guilherme Bezerra Alves Instituto de Medicina Integral Prof. Fernando Figueira (IMIP) Brazil
REVIEW RETURNED	19-Jan-2021
GENERAL COMMENTS	All of my comments have been properly followed